# XANES Measurements for Studies of Adsorbed Protein Layers at Liquid Interfaces

**DOI:** 10.3390/ma13204635

**Published:** 2020-10-17

**Authors:** Oleg V. Konovalov, Natalia N. Novikova, Mikhail V. Kovalchuk, Galina E. Yalovega, Alexey F. Topunov, Olga V. Kosmachevskaya, Eleonora A. Yurieva, Alexander V. Rogachev, Alexander L. Trigub, Maria A. Kremennaya, Valentin I. Borshchevskiy, Daniil D. Vakhrameev, Sergey N. Yakunin

**Affiliations:** 1European Synchrotron Radiation Facility, 38043 Grenoble, France; konovalo@esrf.fr; 2National Research Center “Kurchatov Institute”, 123182 Moscow, Russia; nn-novikova07@yandex.ru (N.N.N.); koval@nrcki.ru (M.V.K.); a.v.rogachev@ya.ru (A.V.R.); alexander.trigub@gmail.com (A.L.T.); s.n.yakunin@gmail.com (S.N.Y.); 3Faculty of Physics, Southern Federal University, 344006 Rostov-on-Don, Russia; galayalovega@gmail.com (G.E.Y.); kremennayamariya@gmail.com (M.A.K.); 4Bach Institute of Biochemistry, Federal Research Center of Biotechnology, Russian Academy of Sciences, 119071 Moscow, Russia; rizobium@yandex.ru; 5Veltischev Research and Clinical Institute for Paediatrics, Pirogov Russian National Research Medical University, 117997 Moscow, Russia; ea-yurieva37@yandex.ru; 6Moscow Institute of Physics and Technology, Dolgoprudny, 141701 Moscow Region, Russia; borshchevskiy@gmail.com (V.I.B.); vakhrameev.dd@phystech.edu (D.D.V.)

**Keywords:** metalloproteins, XANES, zinc binding sites, protein layers at liquid interface, Langmuir trough

## Abstract

X-ray absorption near edge structure (XANES) spectra for protein layers adsorbed at liquid interfaces in a Langmuir trough have been recorded for the first time. We studied the parkin protein (so-called E3 ubiquitin ligase), which plays an important role in pathogenesis of Parkinson disease. Parkin contains eight Zn binding sites, consisting of cysteine and histidine residues in a tetracoordinated geometry. Zn K-edge XANES spectra were collected in the following two series: under mild radiation condition of measurements (short exposition time) and with high X-ray radiation load. XANES fingerprint analysis was applied to obtain information on ligand environments around zinc ions. Two types of zinc coordination geometry were identified depending on X-ray radiation load. We found that, under mild conditions, local zinc environment in our parkin preparations was very similar to that identified in hemoglobin, treated with a solution of ZnCl_2_ salt. Under high X-ray radiation load, considerable changes in the zinc site structure were observed; local zinc environment appeared to be almost identical to that defined in Zn-containing enzyme alkaline phosphatase. The formation of a similar metal site in unrelated protein molecules, observed in our experiments, highlights the significance of metal binding templates as essential structural modules in protein macromolecules.

## 1. Introduction

Two-dimensional arrays of proteins adsorbed at lipid membranes in living organisms have attracted exceptional interest in biomedical studies. According to modern conceptions, the real biochemistry processes in a cell proceed not in three-dimensional (3D), but in two-dimensional (2D) (or even one-dimensional (1D)) space. Various micro- and nanostructures in the cell are organized in such a way that proteins and other biomolecules are either permanently bound to their surfaces, or adhere on surfaces, reacting with different molecular species. As a result of such interactions, the physico-chemical parameters of protein macromolecules, as well as the general mechanism of action, can significantly alter.

High precision X-ray techniques have been recognized to be very promising for investigations of two-dimensional bioorganic nanostructures. Nowadays, X-ray reflectometry, grazing incidence diffraction, grazing incidence small-angle scattering, the X-ray standing wave method, etc. are extensively used to study adsorbed protein layers at fluid interfaces [1,2,3]. However, these techniques do not provide atomically resolved structural information about protein macromolecules.

X-ray absorption spectroscopy (XAS) expands opportunities to characterize the local structure of the absorbing sites in biological materials. Analysis of XAS spectra for metalloproteins and metal-protein complexes allows one to obtain detailed information about the 3D geometry of a metal site, the number and type of ligands, metal-ligand distances, and the oxidation state of metal ions [4,5,6,7,8]. Metalloproteins play a key role in many vital biological processes, such as enzymatic catalysis, gas transport, electron transfer, and redox signal transmission. Transition metals (Zn, Cu, Ni, Co, Fe, and Mn) in proteins coordinate with a wide range of ligand systems and are involved in the regulation of the reactivity of active centers in metalloproteins by modulating its electronic structure, redox potential, steric factor, dielectric properties, etc. Even subtle changes in the structural parameters of the metal-ligand system can induce severe alterations of the biological functions of metalloproteins.

In recent years, much attention has been paid to metal complexes that have occurred at the interface between biological molecules, with special regard for the role that metal ions play in formation of supramolecular protein ensembles. Numerous investigations have demonstrated that metal ions act as linkage agents for protein macromolecules facilitating the assembling of protein structures with very specific properties, such as high adhesion, rigidity, abrasion resistance, and self-healing [9,10].

For a long time, XAS has been successfully used to study protein macromolecules both in protein crystals and in protein solutions. X-ray absorption near edge structure (XANES) measurements in the fluorescent mode under total external reflection conditions at liquid interfaces, implemented at European Synchrotron Radiation Facility (ESRF) on the ID10 beamline, for the first time and presented in this work, significantly extend the general capabilities of X-ray structural methods. These experiments combine several benefits important for biological studies, i.e., two-dimensional arrays of proteins can be investigated directly at liquid interface and structural changes in reacting protein molecules can be monitored in real-time under physiological conditions.

XANES spectra at liquid interfaces were first obtained by Wang et al [11]. These studies were focused on the interaction between the lipid monolayer, deposited on the water surface in a Langmuir trough, and metal ions (iron) present in the water subphase. In the present work we applied XANES measurement for the characterization of the local environment around metal ions in protein molecules adsorbed at liquid surface in a Langmuir trough. Our object was the parkin protein. Parkin is traditionally mentioned as E3 ubiquitin ligase, but its official name, according to enzyme nomenclature, is RING-type (really interesting new gene) ubiquitin transferase (EC 2.3.2.27). Thus, it belongs to a ssecond class of enzymes, transferases, which catalyze transfer of specific functional groups from the donor molecule to another acceptor molecule. Parkin catalyzes the transfer of ubiquitin residue (ubiquitinyl) from an E2 enzyme conjugated with ubiquitin to an acceptor protein. E1, E2, and E3 are the conventional names of enzymes catalyzing different stages of protein ubiquitination, i.e., the process of marking proteins for proteasome degradation. Parkin is an E3 enzyme and provides the substrate specificity of the ubiquitin-proteasome system [12] and plays a central role in proteasome-mediated degradation of misfolded proteins in dopaminergic neurons.

Parkin belongs to a group of Zn-finger proteins and contains eight Zn binding clusters in four RING domains. Alterations in the structure of parkin are caused by mutations in the gene encoding the protein or posttranslational modifications, and have been shown to be a predominant cause of the accumulation of protein aggregates (mostly the presynaptic protein alpha-synuclein) in the neuron cytoplasm that has a significant implication in the pathogenesis of Parkinson disease. Mutations of cysteine (Cys) residues that disrupt zinc coordination in parkin RING domains have been demonstrated to correlate to the loss of the ligase activity of parkin [13]. These findings highlighted the particular relevance of Zn binding clusters for parkin functioning and have attracted great interest in investigating possible mechanisms responsible for destabilization of the zinc ligand environment in the parkin protein.

## 2. Materials and Methods 

### 2.1. Sample Preparation

The synthetic gene prkn2_rat (GeneArt) was cloned into pMK (kanR) using SacI and KpnI cloning sites. The full-length gene was cleaved by BamHI and XhoI restriction sites and ligated into pGEX-6P1 vector. The recombinant proteins of construct were overexpressed in E. coli KRX cells and grown at 37 °C in lysogeny broth medium containing 100 mg/l ampicillin until the A_600_ level of 0.6 was recorded. Proteins expressions were induced by addition of 0.7 mM IPTG, supplemented with 0.5 mM ZnCl_2_. Following induction, cells were grown overnight at 16 °C. The cells were harvested and lysed by sonication on ice in buffer A (20 mM Tris-HCl (pH 7.4), 120 mM NaCl) supplemented with 1% Triton X-100 and 2 mM PMSF. 

The cell lysates were centrifuged at 48,000 g for 30 min, at 4 °C. The supernatant of GST-parkin was purified on a 20 ml GST column and eluted with 10 mM reduced glutathione. The pull-down fractions were concentrated using Amicon Ultra concentrators with 10 kDa Millipore membrane and separated by gel-filtration on Superdex200 10/300 (Cytiva, Uppsala, Sweden). Purified on affinity chromatography, parkin was eluted from size-exclusion Superdex200 column in V_0_ as oligomers with molecular weight not less than 1000 kDa.

### 2.2. Protein Film Formation

Four hundred microliters of protein solution in glycerol (concentration 1 mg/mL) was spread using a chromatography syringe on the surface of glycerol/water (1/1) subphase in a Langmuir trough. The protein layer was not compressed; however, the surface pressure gradually increased and reached the maximum value π ~ 9–11 mN/m in about 40 min. We started X-ray measurements 1 h after spreading the protein solution in the Langmuir trough.

### 2.3. X-ray Absorption Near Edge Structure (XANES) Measurements

The XANES spectra were acquired at ESRF, at the EH1 end-station for the ID10 beamline (Grenoble, France). This end-station, equipped with the double crystal beam deflector and the home-built Langmuir trough, is devoted for studies on liquid surfaces and interfaces [14]. Monochromatic X-ray beam with flux of 4.4 × 10^11^ ph/s, and cross section 350 × 20 micron^2^ (H × V) illuminated the sample surface area of 4 mm^2^ at a grazing angle 0.1 deg. Experimental setup of the beamline optics is sketched in Figure 1. Three undulators set at energies 9.66, 9.76, and 9.86 keV created an X-ray source with the band width sufficient to measure the XANES spectrum in the range of Zn K-edge. The energy was calibrated by referencing to the absorption edge of a metallic Zn foil, such that the maximum of the first derivative was set at 9659 eV. Energy scan of the downstream Si (111) channel cut monochromator provided the incident beam on the sample with 1.3 eV energy resolution. Afterwards the highly monochromatic X-ray beam was focused with a set of compound refractive lenses on the sample and further deflected to the liquid surface with a pair of flat mirrors (Figure 1).

These mirrors provided the grazing angle of the X-ray beam on the liquid at 90% of the critical angle of total external reflection. Finally, the Vortex EM detector, mounted above the aqueous surface, collected the fluorescence radiation for XANES measurements (Figure 1). This experimental setup was used, for the first time, on the ID10 beamline to perform the pilot XANES experiments on the protein films at the liquid surface. The sealed Langmuir trough was filed with water vapor saturated helium to decrease X-ray scattering on the nitrogen and oxygen contained in the air, and to reduce evaporation of the liquid. All XANES data were collected at room temperature (T = 23 °C).

## 3. Results

### 3.1. Investigations of Protein Arrangement at the Air/Liquid Interface

We applied the X-ray standing wave (XRSW) method to examine the protein layer formed at the air/liquid interface in Langmuir trough. The XRSW technique is based on the analysis of angular dependence of secondary radiation (e.g., characteristic fluorescence), exited by the incident X-ray beam, which allows one to determine the position of atoms emitting secondary radiation signal. XRSW studies under total external reflection have been demonstrated to be very informative for characterization of thin bioorganic films particularly at liquid interfaces [3]. 

Experimental measurements were carried out on the LANGMUIR beamline, Kurchatov Center for synchrotron radiation (Moscow, Russia). The energy of the incident beam was 13 keV. The beam flux was 3 × 10^8^ ph/sec, beam size was 100 micron × 3 mm. The fluorescent signal was measured by an energy dispersive Vortex EX detector mounted normal to the liquid surface. The characteristic fluorescence spectra were recorded for each angle of incidence in the angular range corresponding to the Total external reflection (TER) region. 

Figure 2 shows the experimental angular dependence of integrated intensity for Zn Kα peak. Analysis of these XRSW data provides direct information on the distribution of zinc ions along the normal to the liquid surface (in the direction normal to the film surface), which, in turn, allowed us to locate the parkin molecules. 

The numerical fitting of the Zn-fluorescence curve was performed using recursion formalism developed by Parratt [15]. Good agreement between the experimental data and the model calculations was obtained for two different types of zinc ion distribution. In the case of the one-layer model, zinc ions are assumed to be distributed in the thick layer, which is located directly at the air/liquid interface. In the case of the two-layer model, zinc ions are present in the thin layer, which is located at a certain distance from the air/liquid interface. 

Curve 1, in Figure 2, represents the best fit, obtained in the frame of the one-layer model. According to these results, zinc ions are distributed in the layer with a thickness of 85 ± 7 Å. Through a comparison of this value with the size of the parkin molecule that is evaluated as 91 × 41 × 54 Å^3^ (see insert in Figure 2), we suggest that parkin molecules are arranged as a monolayer underneath the air/liquid interface and the major axis of the protein molecule is directed perpendicular to the liquid surface. However, one cannot exclude the possibility that protein layer is composed of double molecule aggregates with the major axis of the protein molecule oriented parallel to the liquid surface. The best fit for the two-layer model (Curve 2 in Figure 2) corresponds to the following parameters: thickness of the top layer (Zn free) is 60 ± 5 Å and the zinc-containing bottom layer appeared to be rather thin, only 14 ± 3 Å. A possible interpretation of these fitting results is that zinc ions are bound at particular sites in the parkin molecules and that these molecules are arranged as a monolayer with the major axis of the protein directed perpendicular to the liquid surface.

Summarizing the obtained XRSW results, we can deduce that parkin molecules are arranged as a thin film at the air/liquid interface, either as a monolayer (most probable arrangement) or as a double-molecule thick layer.

### 3.2. XANES Measurements for Studying Protein Layer at the Air/Liquid Interface

The XANES spectra for the parkin protein molecules adsorbed at the air/liquid interface in a Langmuir trough were recorded in two series of measurements.

#### 3.2.1. Series I

The first series was carried out with the short time of X-ray exposure. The XANES data were acquired at the Zn K-edge over the range of 300 eV during 750 s. Then, the protein film was left to restore for 40 min. After that, we collected the next XANES spectrum. A total of four XANES spectra were recorded. No changes in these spectra were observed.

The correct calculation of absorbed X-ray radiation dose in experiments at liquid interfaces in total external reflection geometry is a rather complicated task. Recently, Brooks-Bartlett et. al. [16] proposed a simple method for assessing the threshold dose of protein samples in SAXS experiments. As emphasized in [16], “a more robust indicator for radiation damage can be defined as the point whereby three consecutive frames were assessed as being dissimilar to the first frame”. According to these criteria, in the first series of our measurements, the protein layer remained undamaged during XANES data collection, therefore, these experimental conditions can be referred to as “mild”. As a crosscheck, we performed XANES measurements for parkin monolayer at the LANGMUIR beamline (Kurchatov Center for synchrotron radiation). The shape of these XANES spectra appeared to be nearly identical to those recorded by the ID10 beamline (ESRF) under “mild” conditions (see Appendix A).

The experimental and theoretical Zn K-edge XANES spectra are shown in Figure 3. Model calculations were performed in the frame of the finite difference approach implemented in FDMNES code [17]. 

Parkin consists of the following four RING domains: RING1, inBetweenRING (IBR), RING2, and RING0 [17]. Each RING domain has a specific Zn binding sequence that coordinates two zinc ions. All eight Zn binding clusters are built in a tetracoordinated geometry, i.e., in six sites zinc is bound to four Cys residues and in two sites zinc is coordinated to three Cys and one histidine (His) residues. The total theoretical XANES spectrum for parkin was calculated as the weighted sum of spectra for two non-equivalent positions in crystallographic structure (PDB entry 4K95) [18]. The calculations were carried out for a full potential; a cluster radius of 7 Å was used. 

As clearly seen in Figure 3, theoretical spectrum is inconsistent with experimental XANES data, obtained in our experiments. The most evident differences concern the white line features. This observation implies that local zinc environment in our parkin preparations differs from that in Zn binding sites defined in parkin by X-ray diffraction studies.

To interpret experimental XANES spectra, we applied qualitative method, so-called “XANES fingerprinting”. A few XANES spectra from mononuclear four-coordinated zinc binding sites available in the literature [19,20,21,22] are presented in Figure 3. On the basis of a comparison of these experimental spectra, two types of Zn binding site are very close to that formed in the studied parkin preparations, 2His1Cys and 2His1Glu. Furthermore, the XANES spectra for parkin appeared to be similar to those obtained by our group in [22] for a monolayer of hemoglobin treated with a solution of ZnCl_2_ salt. The ligand environment around zinc ions in hemoglobin has been studied in [23]. Zinc ions were demonstrated to be coordinated to at least three amino acids, i.e., His146β, Cys93β and His143β [23,24]. Thus, XANES data obtained in our experiment can be attributed to the presence of two histidine and one cysteine/glutamate residues in the ligand environment of zinc ions bound in parkin molecules. 

None of the eight Zn binding sites in parkin RING domains (PDB entry 4K95) contains two histidine residues. Assuming zinc ions are coordinated by two histidine and one cysteine ligands, we can define parkin polypeptide chain closely located at Cys253, His255, and His257 which might form resembling binding sites. Therefore, we hypothesize that parkin was not properly folded during expression/purification procedures and its native Zn binding sites were not formed. However, Cys253/His255/His257 was still available for Zn binding even in the misfolded protein due to a proximity in protein sequence.

#### 3.2.2. Series II

The second series of measurements was focused on the evolution of local environment around zinc ions under increased X-ray radiation load. In this series, XANES spectra were recorded one after the other (without any pause). Total acquisition time of each spectrum was 2250 s. We examined four different preparations of the parkin protein. A total of five spectra were recorded for each preparation. Figure 4 shows the first (Curve 2) and the last (Curve 3) XANES spectra collected for one of the studied preparations. Characteristic changes in these experimental data, that are clearly seen in this Figure, can be attributed to the progression of radiation damage.

Even qualitative comparison of XANES experimental data obtained in the first and second series evidenced the pronounced effect of X-ray radiation on the local structural environment of zinc ions in the studied parkin preparations. Most noteworthy, in Figure 4, is the increase in the height of the white line in XANES spectra recorded in the second series. This change is remarkable, indicating the formation of a Zn binding site with different ligand geometry. Indeed, as has been shown in [19,20] the intensity of the white line in XANES spectra for Zn-containing proteins is in strong correlation with zinc coordination number. Tetracoordinated Zn complexes are characterized by a low intensity of the white line, less than 1.5 (XANES spectra are normalized to a unit edge jump); in the case of penta- or hexa-coordination, the intensity of the white line is higher and never falls below 1.6. Following these criteria, we assume that in parkin preparations, exposed to high X-ray radiation load, zinc is coordinated to five or six ligands.

Another clear distinction between experimental data, shown in Figure 4, is a slight shift in the energy position of the first minimum (marked by the arrow) toward the smaller energy values on XANES spectra, recorded in the second series. These transformations can be attributed to the increase in the Zn-ligand distances in parkin preparations [25], which seems quite reasonable by taking into account the increase in zinc coordination number. Note, the elongation of zinc-ligand bond length with an increase in the number of binding ligands, observed in our experiments, is consistent with findings reported in [26].

XANES fingerprint analysis (Figure 5) shows that the first experimental spectrum for parkin, collected at high X-ray radiation load, practically coincides with that for a monolayer of Zn-containing enzyme alkaline phosphatase [27].

The crystallographic structure of swine (*Sus scrofa* L) alkaline phosphatase (EC 3.1.3.1), studied in [27], is not known. However, the coordination spheres of zinc ions may be deduced by similarity with other organisms, for instance, with alkaline phosphatase of other organisms, for instance, human one (PDB entry 3MK2). According to X-ray diffraction data, in human alkaline phosphatase, each zinc ion is coordinated to six residues. Essentially, the binding of zinc ions in alkaline phosphatase occurs through aspartate, histidine, and “non-standard” phosphoserine residues, no cysteine is present in local zinc environment. At the same time, Cys residues are among the most sensitive to X-ray modifications amino acids [28]. We suppose that under increased X-ray radiation load Cys residues were the first to suffer from X-ray radiation and were removed from Zn-coordination shell. Such changes in parkin will result in XANES spectra similar to alkaline phosphatase with Cys-free Zn coordination.

It should be noted that despite these alterations of local environment in the studied parkin preparations, Zn binding sites remained structurally ordered under the conditions of our experiments.

## 4. Discussion

Zinc belongs to the group of most abundant transition metal in protein macromolecules. Over 10% of human proteins contain zinc [29]. Zinc is known to play a key role in stabilizing the protein structure and to exert the stabilization effect at all levels of protein structure organization, i.e., secondary, tertiary, and quaternary. In most proteins (~90%) zinc ions are coordinated to cysteine, histidine, aspartate, and glutamate residues [29,30]. Depending on their functional role, zinc binding sites in protein can be divided into the following five main classes: catalytic, structural, cluster, transport, and intermolecular [30,31,32]. The most common are structural sites consisting of Cys and His residues in a tetrahedral geometry [29,30]. Cysteine is the preferential amino acid in such sites.

In the present studies, we investigated the parkin protein that belongs to the Zn-finger family and contains eight tetracoordinated Zn binding sites. On the basis of the obtained XANES data, we identified two types of local zinc environment in our parkin preparations depending on X-ray radiation load. According to our experimental results, under mild conditions, the zinc sites are four-coordinated sites; but it appeared that local zinc environment does not correspond to that determined in parkin crystallographically. Essentially, we found that zinc site structure in our parkin preparations was very similar to that identified in hemoglobin, treated with a solution of ZnCl_2_ salt. Under high X-ray radiation load, considerable changes in the zinc site structure were detected; zinc coordination number increased to six, while the ligand environment became almost identical to that defined in Zn-containing enzyme alkaline phosphatase.

The biological reasons for occurrence of similar metal binding sites in metalloproteins have been actively discussed in the past years. Structurally conservative binding sites in Zn- and Ca-containing proteins were investigated in [33]. Torrance et al. checked the Protein Data Bank and used the protein structure comparison methods to search the cases of structural templates with the same number of liganding residues in similar geometry. It has been demonstrated that the structure of Zn binding sites can differ greatly in a group of related proteins (with high level of sequence identity) and be very similar in unrelated proteins that have completely different conformations. These sites have independently evolved in a large number of proteins, indicating that unrelated proteins use the same set of residues to bind metals.

Recently, Rosato et al. proposed the concept of minimal functional sites in order to describe structure-function relations in metalloproteins [34]. Minimal functional sites were defined as 3D structures with conserved geometry that include the nearby region around the metal site and did not depend on the macromolecular structure of the protein. Structural bioinformatics analysis demonstrated, that equivalent minimal functional sites can occur in significantly divergent proteins. Generally, Rosato et al. defined minimal functional sites as essential structural units, that are “grafted onto the protein fold” and play an important role in maintaining protein functions. The close similarity between Zn binding sites in the studied parkin preparations and two unrelated proteins (hemoglobin and alkaline phosphatase), observed in our experiments, is consistent with the concepts mentioned above.

Remarkable is the fact that specific Zn binding templates with distinct topologies in our parkin preparations occurred as a result of random changes in protein conformation (due to misfolding or X-ray radiation damage). These findings highlight the biological relevance of metal binding templates and are in the mainstream of the general idea that tends to specify metal sites as substantive elements in protein macromolecules. 

In conclusion, the presented results clearly demonstrated the great potential of XANES measurements at liquid surface for structural studies of biological materials. Due to a drastic decrease in background scattering intensity under total external reflection conditions (accordingly a considerable increase in the signal-to-noise ratio) low spectroscopic signal from metal atoms bound in trace amounts by protein molecules can be detected in such experiments. Concentration of protein molecules at the liquid interface due to self-assembling is another important experimental advantage, enabling XANES spectra to be collected for a single-molecule thick protein layer. Additionally, the possibility to examine non-crystalline protein systems under nearly physiological conditions is critical in biomedical researches. All the mentioned benefits are extremely challenging for studying biophysical properties of proteins and understanding their functions in complex cellular processes.

## Figures and Tables

**Figure 1 materials-13-04635-f001:**
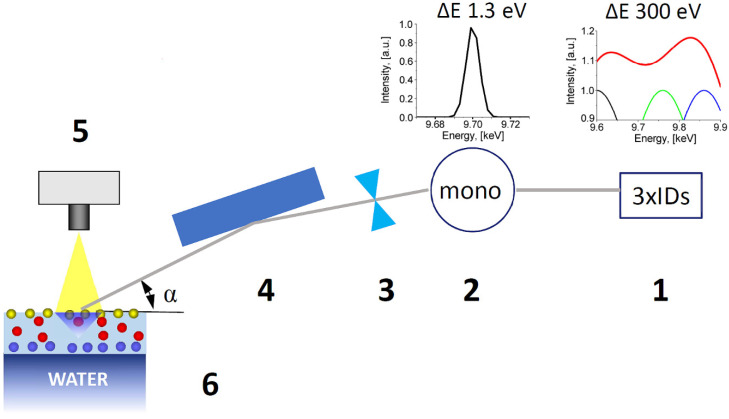
Schematic view of the X-ray absorption near edge structure (XANES) setup of the ID10 beamline at European Synchrotron Radiation Facility (ESRF). (**1**) Three undulators and corresponding energy spectrum produced by them; (**2**) Si(111) channel cut monochromator and the energy spectrum after it; (**3**) Compound refractive Be lenses for beam focusing on the sample; (**4**) Flat mirror deflecting the X-ray beam to the liquid surface below the critical of total reflection; (**5**) Energy dispersive Vortex EM detector; (**6**) Langmuir trough with the sample.

**Figure 2 materials-13-04635-f002:**
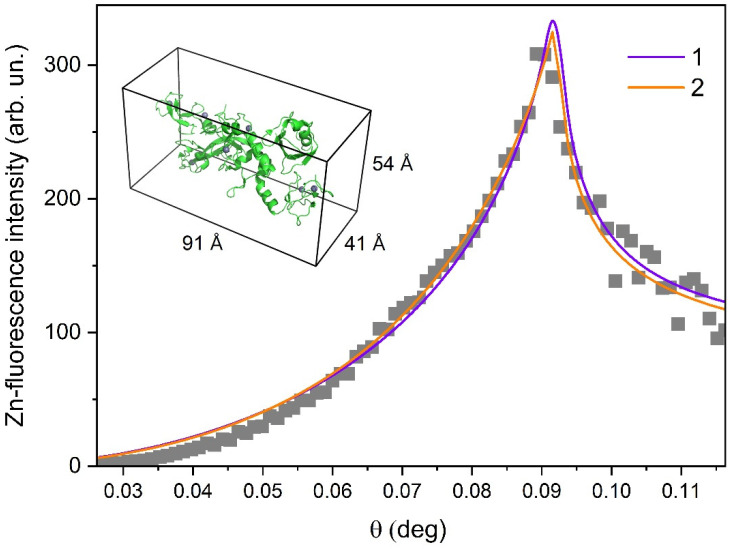
Experimental angular dependence of the Zn-fluorescence yield from layer of the parkin protein. The solid curves represent the angular dependences, calculated for one-layer (Curve 1) and two-layer (Curve 2) models. The three-dimensional (3D) structure of the parkin protein is shown in the insert.

**Figure 3 materials-13-04635-f003:**
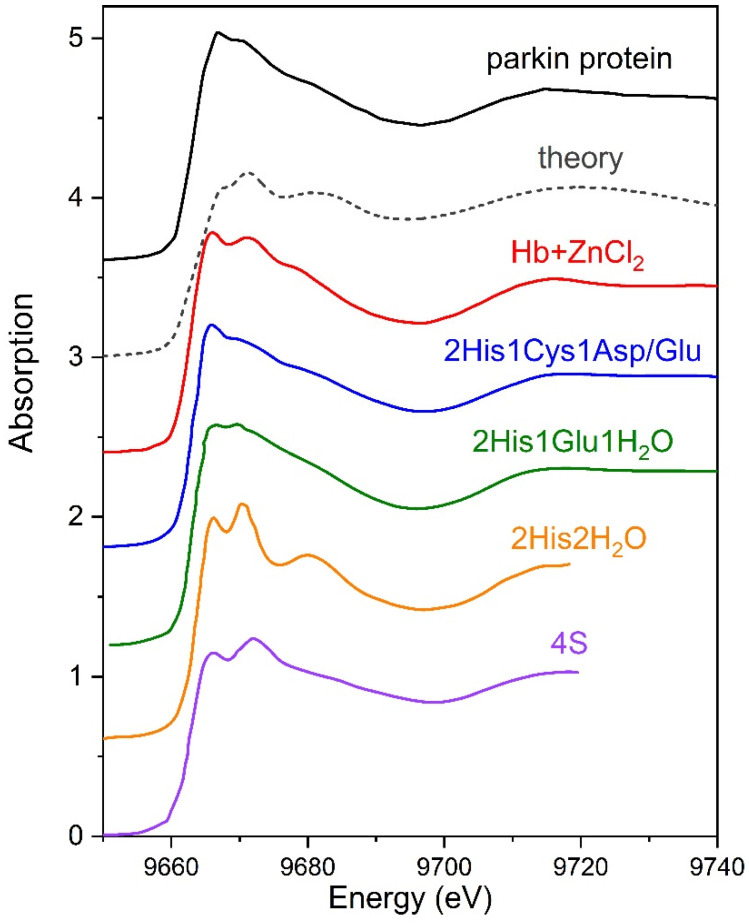
Comparison of XANES spectra at the Zn K-edge for the parkin protein layer adsorbed at the liquid interface and experimental data available in the literature [19,20,21,22]. Dashed line, calculations. For clarity, the curves are offset vertically.

**Figure 4 materials-13-04635-f004:**
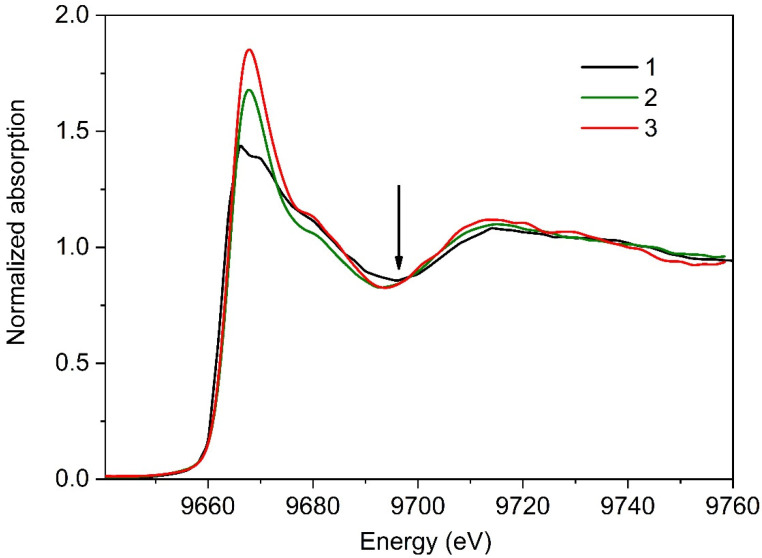
Zn K-edge XANES spectra for the parkin protein layer adsorbed at liquid interface, collected under mild conditions of measurements (Curve 1) and under high X-ray radiation load (Curves 2 and 3).

**Figure 5 materials-13-04635-f005:**
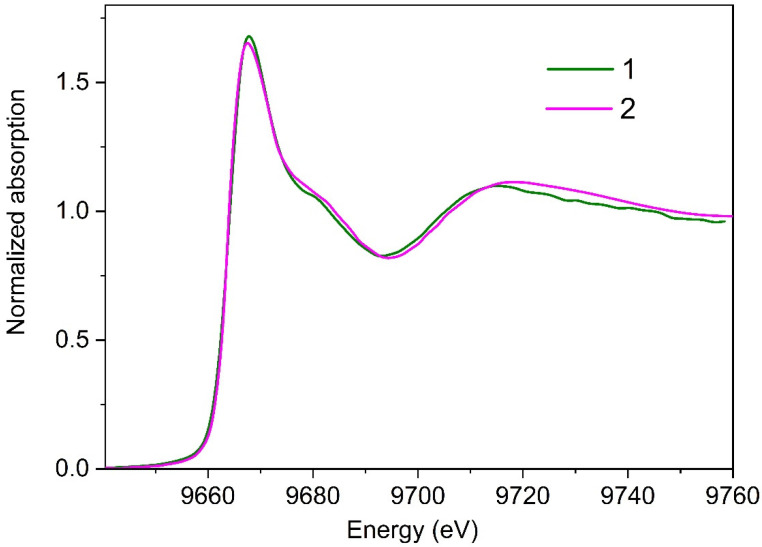
Zn K-edge XANES spectra for the parkin protein layer adsorbed at liquid interface, collected under high X-ray radiation load (Curve 1), and for the monolayer of Zn-containing enzyme alkaline phosphatase (Curve 2).

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
