# Peer review of "XANES Measurements for Studies of Adsorbed Protein Layers at Liquid Interfaces"

_materials, 2020, doi:10.3390/ma13204635_

Round 1

Reviewer 1 Report

This study investigated XANES measurement of parkin protein layers adsorbed at liquid layer. Since protein layers are formed at the liquid interface, the parkin protein is in the real environment. Zn K-edge XANES clearly indicated different types of Zn environment exists in the parkin protein. It is also interesting that the structures of parkin protein is changed with the dose of X-ray. I think this manuscript is suitable for Materials with the minor revisions described below.

(1) Line 193. What is the photon flux of X-rays during the X-ray radiation load? It is important for the discussion of the X-ray radiation damage.

(2) Line 259. It is interesting that the Zinc site structure in the parkin protein is close to the hemoglobin. Is it possible to discuss this reason?

(3) Line 265. It is better to discuss the proposed mechanism of structural changes of the parkin protein with the X-ray radiation load.

(4) It is minor correction. Line 149 should be 3.1. Series I. Line 191 should be 3.2. Series II.

Reviewer 2 Report

Please see the attach file.

Round 2

Reviewer 2 Report

The reviewer appreciates the authors' comments, corrections and details added in the MS. The authors have addressed all of my major concerns. I found the XRSW measurements present in the MS really interesting.

I have just one thing to say (but it's a detail and it doesn't change the results): to reinforce the "mild aspect" of the first series, the comments given in the response ("Moreover we performed additional XANES measurements on parkin monolayer at LANGMUIR beamline, Kurchatov center for synchrotron radiation (Moscow, Russia) with the beam flux of 3e+8 ph/s. The shape of these XANES spectra was identical to those recorded at ESRF under “mild” conditions") can be given in the MS, as it is an unquestionable evidence.
